# Genetic, Epigenetic, and Non-Genetic Factors in Testicular Dysgenesis Syndrome: A Narrative Review

**DOI:** 10.3390/genes17010040

**Published:** 2025-12-31

**Authors:** Alessandro Ciarloni, Nicola delli Muti, Sara Sacco, Nicola Ambo, Valentina Di Giacomi, Michele Perrone, Silvia Rossi, Marinella Balercia, Gianmaria Salvio, Giancarlo Balercia

**Affiliations:** 1Endocrinology Clinic, Department of Clinical and Molecular Sciences, Polytechnic University of Marche, 60126 Ancona, Italyn.dellimuti@staff.univpm.it (N.d.M.); s.sacco@staff.univpm.it (S.S.);; 2CDL Biomedical Laboratory Techniques, DADP Degree Courses, Azienda Ospedaliero Universitaria delle Marche, Faculty of Medicine and Surgery, Polytechnic University of Marche, 60126 Ancona, Italy

**Keywords:** endocrine disruptors, testicular dysgenesis syndrome, cryptorchidism, hypospadias, testis cancer, anogenital distance

## Abstract

Background: Testicular dysgenesis syndrome (TDS) is a complex disorder of the male reproductive system related to disfunction of the fetal testis. The clinical features of TDS may be evident at birth or infancy (cryptorchidism, hypospadias and/or reduced anogenital distance) or occur later in adulthood (testis cancer, infertility). Genetic background seems to be important for genetic predisposition, with new genes being associated with components of the syndrome in last years. Interestingly, the incidence of clinical manifestations of TDS has been increasing in many countries in recent decades, suggesting that genetic predisposition alone cannot explain this trend. Consequently, the hypothesis of multifactorial etiopathogenesis is becoming increasingly accepted nowadays, with environmental factors probably acting during early developmental stages in genetically predisposed individuals. Methods: In this narrative review, we aim to critically evaluate genetic and non-genetic factors involved in the pathogenesis of TDs. Results: Important associations with intrauterine growth disorders and maternal diseases (overweight/obesity and diabetes) as well as lifestyle factors (e.g., smoking and alcohol abuse) were found. In such context, endocrine disruptors probably play a major role. These substances are widely used in industry and can exert estrogenic and antiandrogenic effects, potentially interfering with the development of the fetal gonad. Conclusions: Considering their possible impact on male sexual health, more attention should be focused on maternal modifiable factors to confirm with prospective studies the mixed results of available evidence.

## 1. Introduction

Testicular dysgenesis syndrome (TDS) is characterized by dysfunction of the fetal male genitals leading to various clinical features that can manifest at birth or later in life [1]. The most common components are male infertility, hypospadias, cryptorchidism, and testicular germ cells tumors (GCTs) [2]. Recent evidence suggests that a shorter anogenital distance (AGD) could also be part of this syndromic picture [3]. Despite the association between cryptorchidism and testicular cancer being noted since the 1930s [4], a possible correlation with other conditions was only proposed after several years (Figure 1). Indeed, the first histological finding suggesting an association between male infertility and testicular cancer dates 1972 [5]. On the other hand, an increased risk of testicular cancer in patients affected by hypospadias was first observed in a Danish cohort study published in 1996 [6] and then confirmed in other large population-based studies [7,8]. Considering the co-occurrence of these conditions and the epidemiological overlap, in 2001, Skakkebaek et al. [9] suggested a common etiopathogenesis in the context of TDS. The concept of TDS is somewhat similar to that of disorders of sex development (DSDs), a group of congenital conditions involving atypical development of chromosomal, gonadal, or anatomical sex [10]. Analogies with subcategory 46, XY DSD, referring to XY karyotype patients, can also be found. A distinction from complete gonadal dysgenesis is easy to carry out considering the female phenotype. Instead, the clinical picture of partial gonadal dysgenesis, characterized by mild-to-severe penoscrotal hypospadias and reduced sperm production, can often mimic TDS. However, the phenotype of TDS is usually less pronounced than that of DSD, probably because of the stronger impact of genetics on the pathogenesis of DSDs [11,12]. In fact, even if nowadays the existence of TDS is strongly supported by the literature [2], its etiology remains unclear. Genetics play a central role, as suggested by the association between genetic alterations and all the components of TDS [13,14,15,16]. In more detail, genetics could be particularly relevant for male infertility, since many cases of idiopathic infertility may be linked to unknown genetic causes [17,18]. However, genetics does not seem to be the only factor contributing to the origin of TDS. Indeed, many studies have reported an increasing incidence of TDS cases, especially in developed countries [19]. Many chemical substances, so-called endocrine disruptors, have shown endocrine activities like estrogenic or antiandrogenic effects potentially interfering with the development of fetal male genitals [2,20,21,22]. Nowadays, there are more than 800 substances that can be classified as endocrine disruptors [23]. These substances are mainly found in plastics, building materials, combustion products, and pesticides, as well as in cosmetics and some foods [24,25,26,27,28,29].They are now practically ubiquitous, being commonly found in air and water [29,30]. Therefore, pregnant women may encounter these substances, which can then affect fetal development [31,32]. In addition, maternal lifestyle can represent an additional critical factor, since cigarette smoking, alcohol, obesity, and diabetes have been associated with the occurrence of TDS [33,34,35]. All these potential contributors to TDS should also be evaluated considering the declining trend in semen quality [36] and serum testosterone levels worldwide [37,38], with associated health issues [39]. Indeed, this decline could also be related to endocrine disruptors [40,41] and the increasing incidence of obesity and metabolic syndrome correlated with unhealthy eating habits [42,43,44].

In this narrative review, we aim to critically evaluate genetic and non-genetic factors involved in the pathogenesis of TDS, and to assess possible future interventions to mitigate the increasing incidence of TDS.

## 2. Materials and Methods

An extensive literature search was conducted using Scopus and PubMed databases up to 30 June 2025. Only articles written in English were included, with no restrictions on publication date or study design.

The search terms included “testicular dysgenesis syndrome”, “testicular cancer”, “cryptorchidism”, “hypospadias”, “male infertility”, “couple infertility”, “anogenital distance”, “pregnancy”, “diabetes”, “overweight”, “obesity”, “cigarette”, “smoking”, “alcohol”, “coffee”, “diet”, “assisted reproductive techniques”, “drugs”, “genetics”, “epigenetics”, “endocrine disruptors”, “pollution”, “chemicals”, and “androgen receptor”. Boolean operators (AND/OR) were applied to create various combinations between search terms. References from retrieved studies were searched for further relevant literature.

To ensure transparency and consistency in our narrative review, we strictly adhered to the SANRA assessment scale (Scale for the Assessment of Non-Systematic Review Articles) [45], a 7-item tool that evaluates (1) justification of the article’s importance for the readership, (2) statement of concrete aims or formulation of questions, (3), description of the literature search, (4) referencing, (5) scientific reasoning, and (6) appropriate presentation of data.

## 3. Genetic and Epigenetic Causes of TDS

### 3.1. Cryptorchidism

Cryptorchidism is one of the earliest detectable signs of TDS [15], and it is defined as the failure of the testis to descend into the scrotal position [46]. Interestingly, the adult testicles of individuals affected by cryptorchidism show a characteristic organization of Leydig cells into micronodules, with a distribution of Reinke crystals within them that could correlate with the degree of testicular function impairment [47]. In addition, Sertoli cells of subjects with TDS show signs of immaturity similar to those of adults treated with high doses of estrogen and antiandrogens [48]. Testicular descent is a process occurring in two main phases: the transabdominal phase, primarily mediated by Insulin-like factor 3 (INSL3) secreted by fetal Leydig cells, and the inguino-scrotal phase, which is strongly dependent on androgen action and androgen receptor (AR) function. Mutations or polymorphisms in *INSL3* and *RXFP2* genes (encoding the INSL3 receptor) are associated with cryptorchidism [49,50]. However, pathogenic mutations of these genes are rare in isolated cryptorchidism cases, and findings are often inconsistent across different studies and populations [49,50]. Moreover, gene expression studies have shown that the exposure to endocrine-disrupting chemicals (EDCs) can induce *INSL3* downregulation, further highlighting the gene–environment interplay [51]. Beyond *INSL3* and *RXFP2*, other genes involved in testicular development and hormonal regulation—such as *NR5A1* (Steroidogenic Factor 1), *WT1* (Wilms tumor 1), *SOX9* (SRY-Box Transcription Factor 9), *DHH* (Desert Hedgehog), and *GATA4*—are implicated. *NR5A1* mutations are associated with a large spectrum of disorders of sex development (DSDs), including isolated cryptorchidism, male infertility, and hypospadias [52,53,54]. Nevertheless, identified variants often show high heterogeneity and are also detectable in phenotypically normal individuals. Furthermore, the *AR* gene was also studied, particularly in patients with bilateral cryptorchidism and associated hypospadias [55]. Point mutations and polymorphic repeats in the *AR* (e.g., CAG repeat length in the N-terminal domain) were associated with reduced androgen sensitivity during fetal life. However, available data remains inconclusive and often lacks replication in large-scale studies. *KCTD13* (potassium-channel-tetramerization-domain-containing-13) is another candidate gene potentially involved in cryptorchidism and hypospadias. In mice, *KCTD13* haploinsufficiency causes an increased incidence of cryptorchidism. Loss of *KCTD13* in cell lines is associated with decreased intracellular levels of AR [56].

Moreover, epigenetic mechanisms may also contribute to the pathogenesis of cryptorchidism. Studies on testicular tissue of fetuses with undescended testes shown alterations in DNA methylation at regulatory regions of key genes involved in gonadal function [57], and the expression of regulatory microRNAs (miRNAs) involved in androgen signaling or Leydig cell differentiation appears disrupted in pathological samples [58].

In addition to these main genes implicated in testicular descent and differentiation, several other genes were identified through genome-wide association studies (GWAS), transcriptomic analyses, and functional studies in animal models. For instance, *DMRT1* (Doublesex and Mab-3 Related Transcription Factor 1), a key regulator of testis development, was implicated in testicular dysgenesis and germ cell survival [59]. *AXIN1*, involved in Wnt/β-catenin signaling, may influence gonadal development through its role in cellular proliferation and differentiation [60]. Other genes such as *ATRX* (Alpha Thalassemia/Mental Retardation Syndrome X-linked), *PIWIL1* (Piwi Like RNA-Mediated Gene Silencing 1), and *CPEB1* (Cytoplasmic Polyadenylation Element Binding Protein 1) are involved in germ cell maturation and meiosis, processes that may be disrupted in the context of TDS [61,62].

A key factor in understanding the genetics of cryptorchidism is exploring its interaction with the intrauterine environment [63]. In fact, maternal exposure to phthalates, pesticides, cigarette smoke, or obesity has been linked to increased risk of cryptorchidism in male children, likely through interference with hormonal signaling and fetal gene expression [64].

Table 1 summarizes the current evidence available on the role of genetic alterations in the origin of TDS.

### 3.2. Hypospadias

Hypospadias is a common congenital abnormality characterized by incomplete fusion of the urethral folds during embryological development of the penile urethra, leading to a proximally displaced urethral meatus, typically located on the ventral surface of the penis [65]. This condition is considered one of the cardinal features of TDS [66].

The prevalence of hypospadias varies across regions, with the highest rates reported in North America (1 in 140 male boys) [67], and lower frequencies in Europe (1 in ~500) [68].

Nowadays, hypospadias is widely recognized as a heterogeneous condition with multifactorial etiology, influenced by genetic susceptibility, environmental exposures, maternal-placental factors, and ethnic background [69,70].

This condition is thought to arise during a critical developmental window between 8 and 14 weeks of gestation, referred to as the male programming window (MPW). During this time, appropriate androgen exposure is essential for the correct differentiation and virilization of external male genitalia [71]. Inadequate androgen signaling during the MPW is hypothesized to lead to the characteristic phenotypic spectrum of TDS, including hypospadias. Despite its complexity, genes regulating testicular development, steroidogenesis, and androgen signaling—particularly those involved in androgen synthesis, metabolism, and receptor function—play a pivotal role in its pathogenesis, especially when linked to the TDS continuum [16,72].

Among these genes, the AR has been extensively studied. Polymorphic CAG repeats in exon 1 of the AR gene inversely affects receptor transactivation; longer CAG repeats reduce AR activity and were associated with reduced fertility and hypospadias in multiple studies [73,74]. For instance, AR CAG repeat length was significantly greater in 44 boys with isolated hypospadias compared to 79 controls (mean: 24.4 ± 2.8 vs. 22.7 ± 3.3; *p* < 0.05) [75]. Similarly, a meta-analysis of six case–control studies (444 cases vs. 727 controls) also reported a mean increase of +1.36 CAG repeats in hypospadias patients [76], an outcome replicated in a cohort of 211 Caucasian boys [77]. However, not all findings are consistent: one screening found no significant CAG repeat differences in 21 patients, suggesting that this polymorphism may not be universally relevant across all populations or hypospadias types [78].

In addition, other sequences may be involved. A large-scale whole-exome sequencing (WES) study revealed that approximately 27% of severe hypospadias cases carried rare damaging variants in genes critical to testosterone synthesis and signaling, including *AR*, *NR5A1*, and *SRD5A2* [79]. Indeed, *NR5A1* encodes Steroidogenic Factor-1 (SF-1), essential for steroidogenesis in Leydig cells. Loss-of-function mutations impair testosterone biosynthesis, resulting in under-virilization. *NR5A1* mutations were reported in individuals with hypospadias, micropenis, and cryptorchidism, all clinical features that align closely with the TDS phenotype [79,80]. Whereas, *SRD5A2* encodes 5α-reductase type 2, the enzyme responsible for converting testosterone to dihydrotestosterone (DHT), the key androgen for external genitalia development. Mutations in *SRD5A2* result in variable degrees of hypospadias, sometimes with micropenis and undescended testes, still placing these cases within the TDS continuum [79]. Furthermore, rare AR variants were significantly enriched among patients with severe hypospadias in the WES cohort [79]. In addition to genes directly involved in AR structure and function, new candidate genes that regulate AR availability are emerging. Seth et al. identified *KCTD13*, a gene at the 16p11.2 locus, in a significant proportion of patients with genitourinary anomalies including hypospadias, micropenis, and cryptorchidism. *KCTD13* modulates androgen receptor (AR) nuclear localization via ubiquitin ligase activity. In vitro knockdown of *KCTD13* reduces nuclear AR levels without affecting AR expression. In the same way, in vivo *KCTD13*-null mice exhibited reduced nuclear AR levels, decreased expression of *SOX9*, smaller testis, cryptorchidism, micropenis, and subfertility. These findings suggest that *KCTD13* plays a pivotal role in the intracellular handling of AR and its disruption may contribute to hypospadias occurrence by impairing androgen signaling during urogenital development [56].

Alongside these contributors, rare monogenic disorders such as *WT1* mutations also produce TDS-like phenotypes [81]. Although TDS is often regarded as a multifactorial syndrome influenced by polygenic and environmental factors, *WT1* mutations, typically linked to syndromic DSDs, result in gonadal dysgenesis and external genital anomalies, including penoscrotal hypospadias and cryptorchidism, reflecting features of testicular dysgenesis [81]. These cases, although syndromic, demonstrate how disruption of Sertoli and Leydig cell function by a single gene can mirror traits of the TDS spectrum [82,83]. A 2018 comprehensive review by Kalfa et al. emphasized that *WT1*, *SRY*, and members of key developmental pathways including *HOX*, *SHH*, *FGF*, and *WNT* gene families also regulate genital and urethral formation. These genes act upstream or in parallel to androgenic pathways, supporting the concept that early disruptions in gonadal programming, whether monogenic or polygenic, may underlie the clinical and anatomical features of TDS [84].

Beyond genetic variants, there is growing evidence that epigenetic dysregulation, including DNA methylation, histone modification, and non-coding RNA expression, plays a role in hypospadias pathogenesis and reflects a key mechanism in environmentally induced TDS [85].

In 2011, Vottero et al. found increased methylation of the AR promoter in preputial tissue from boys with hypospadias, along with higher levels of a specific DNA methyltransferase (DNMT3A) resulting in decreased gene expression. Crucially, in vitro exposure to dihydrotestosterone (DHT) or testosterone reversed these changes in a dose-dependent manner, suggesting that epigenetic silencing of AR may underlie hypospadias by suppressing androgen responsiveness during urethral differentiation [86].

Likewise, an epigenome-wide association study on preputial tissue identified 25 CpG sites differentially methylated in hypospadias patients. These changes affected genes involved in androgen metabolism (*CYP4A11*, *EPHX1*, *KLK* family), β-catenin/Wnt signaling (*PKP2*, *TNKS*) and developmental pathways (*WDHD1*, *DNM1L*). Notably, this suggests that epigenetic dysregulation may extend beyond AR itself, involving a broader network of genes, crucial for proper male genital development [87]. 

This idea is further supported by a 2023 pilot study that applied Genome-wide Methylated DNA sequencing (MeD-seq) in foreskin tissues from boys with proximal hypospadias. Although some differentially methylated regions (DMRs) were observed, most notably hypermethylation in *LINC00665* and *MAP3K1*, the findings were variable and did not consistently map onto known DSD-related genes [88].

Adding functional insight, a recent pediatric study investigated both DNA methylation and gene expression in foreskin samples from hypospadias patients. The researchers reported decreased *AR* expression along with increased expression of *ESR1*, *FGFR2*, and *BMP7*, involved in estrogen signaling and genital development. Epigenetically, these changes were mirrored by hypermethylation at *ESR1*, *FGFR2*, and *FGF8* loci as well as hypomethylation at *AR*, suggesting a coordinated yet dysregulated epigenetic program influencing multiple hormonal pathways [89].

### 3.3. Testicular Cancer

Testicular cancer is one of the most crucial elements in the definition of TDS originally proposed by Skakkebaek et al. [9] and it was also associated with poorer prognosis in that syndromic context [90,91]. Most of the testicular cancers (about 95%) are germ cells tumors (GCTs), about half of which are seminomas, mainly occurring in young adult men aged between 15 and 44 years [92]. Recently, a trend toward older age at diagnosis of GCT has been observed, with increasing percentage of seminomas over nonseminomas [93,94]. Cryptorchidism and a family history of testicular cancer are two well-known risk factors for testicular cancer [95], underscoring the strong link between this type of malignancy and genetic and/or congenital abnormalities. However, several acquired factors, such as occupational exposure to toxic substances, alcohol, smoking, and infections, can contribute to the onset of testicular cancer, justifying significant epidemiological variability between different geographical regions in developed and developing areas around the world [96]. In addition, it has been hypothesized that the upward trend in TDS may depend mainly on these acquired factors [2].

The recent revision of the World Health Organization (WHO) classification of testicular tumors introduced some relevant changes. GCTs are now divided into tumors derived from germ cell neoplasia in situ (GCNIS) and tumors unrelated to GCNIS [97] (Table 2).

Currently, GNCIS is the preferred term for the testicular pre-invasive lesion first described by Skakkebaek in 1972 as “carcinoma-in situ” (CIS) [98] and renamed “intratubular germ cell neoplasia unclassified” (IGCNU) or “testicular intra-epithelial neoplasia” (TIN) in the 1980s [99]. In postpubertal subjects with diagnosis of testicular cancer, tubules with GCNIS are frequently found in proximity of invasive GCTs and in the contralateral testicle [100], whereas the occurrence of GCNIS before puberty has been described only in subjects with DSD after surgery or gonadectomy [101].

Hoei-Hansen et al. [102] first provided a systematic evaluation of differential gene expression in GCNIS, comparing pathological samples of GCNIS, invasive GCTs, and normal testicular tissue. Interestingly, they demonstrated an overexpression of several genes involved in testicular growth and development, such as Secreted frizzled-related protein 1 (*SFRP1*), Insulin-like growth factor binding protein-6 (*IGFBP6*), histone deacetylase 3 (*HDAC3*), and decorin (*DCN*). Similarly, Palmer et al. [103] showed an overexpression of microRNAs (miRNAs) miR-371~373 and miR-302 cluster, which are considered markers of embryonic stem cells, suggesting the persistence of an embryonic pattern of miRNA expression. This is in accordance with the hypothesis by Skakkebaek et al. [104], who, based on ultrastructural studies and histochemical findings showing common features of GCNIS cells and gonocytes, elaborated the following statements: GCNIS cells are malignant gonocytes, and their transformation into invasive tumors is hormone-dependent; GCNIS gonocytes may regress into totipotent embryonic cells, giving rise to all the types of non seminomatous GCTs; the ability to regress into embryonic cells decreases with age, and GCNIS that cannot regress can give raise to classical seminomas. More recent studies using genome-wide expression profiling confirmed the expression of a high number of genes linked to the stem cell phenotype in GCNIS cells, with 35–50% expressed genes overlapping between GCNIS and embryonic stem cells (ESCs) [105]. Comparing genetic expression profiles between GCNIS and immature germ cells, Sonne et al. confirmed a close resemblance between gonocytes and GCNIS. Interestingly, they also found that GCNIS showed neither overexpression of oncogenes nor under-expression of tumor suppressor genes, leading the authors to hypothesize that GCNIS cells are not malignant cells sensu stricto, but rather arrested gonocytes [106]. Afterwards, this was supported by the identification of a large range of stemness genes (*POU5F1*, *NANOG*, *LIN28-A*, *DPPA4*, *DPPA5*, *KIT*, and *TFAP2C*) expressed by GCNIS cells, suggesting the formation of GCNIS in utero, with subsequent transformation to malignant germ cells during the peri- and post-puberal period due to further genetic and epigenetic modifications [14]. Among genetic modifications, the gain of extra chromosome arm 12p material has been associated with malignant transformation of GCNIS cells, making them gradually independent of the intratubular environment, with consequent extratubular invasion [107].

To sum up, the majority of invasive GCTs derive from GCNIS cells of fetal origin. It has been hypothesized that GCNIS cells are arrested gonocytes whose maturation defect derives from a congenital dysfunction of Sertoli cells and Leydig cells [2]. A potential impairment of spermatogenesis and reduced testosterone production capacity consequently explains the infertility, cryptorchidism, and hypospadias observed in TDS [108]. There is an extreme lack of information on the relationship between TDS and stromal tumors of the sex cords, although there are sporadic reports of tumors of this type in patients with DSD [109] and gonadal dysgenesis such as Klinefelter syndrome [110]. Further studies on the genetics of sex cord stromal tumors could be therefore useful in clarifying their relationship with TDS.

### 3.4. Male Infertility

Infertility is another important component of TDS. Genetic causes of male infertility are frequent findings, and their evaluation is mandatory in cases of azoospermia or severe oligozoospermia [111]. The more common genetic causes of male infertility are Klinefelter syndrome and y chromosome azoospermia factor (AZF) microdeletions, but other important genetic diseases to consider in specific clinical contexts are cystic fibrosis and hypogonadotropic hypogonadism [112]. It is also possible that many idiopathic cases involve genetic alterations that are still unknown, and new genes potentially involved in fertility problems have been continuously discovered in recent years [17,18].

Considering this complex genetic context, separating genetic alterations related to infertility in TDS from the others is not simple. However, there are many genes related to infertility that are also involved in the pathogenesis of other TDS components. In detail, longer CAG repeat polymorphism of the AR is associated with worse semen parameters [113,114]. However, ethnicity may play a significant role, since the results found in a Russian population [114] were not confirmed in an Iranian population [115]. The impact of AR function on male infertility was also strengthened by a finding of decreased immunohistochemical AR expression in testicular tissue of men affected by non-obstructive azoospermia (NOA) [106]. AGD, defined as the distance from the anus to the genitalia, is a sexually dimorphic trait established during the fetal “masculinization programming window” (approximately gestational weeks 8–14 in humans), depending on adequate androgenic signaling [116]. Reduced AGD in males is considered a sensitive biomarker of impaired androgen action in utero. It was also associated with features of TDS, including cryptorchidism, hypospadias, impaired spermatogenesis, and increased risk of testicular cancer [3,117,118,119,120]. Similarly, the ratio of index to ring finger lengths (2D:4D) (a typical indicator of androgen exposure in utero) has been used to explore the relationship between androgenization and reproductive function, but no clear associations were observed [121].

Many studies, particularly in animal models, highlighted the significant role of genetics demonstrating the partial heritability of AGD and strengthening potential correlations with other aspects of TDS [122,123].

In humans, Sathyanarayana et al. (2012) investigated how polymorphisms in genes known to cause major genitourinary abnormalities could result in borderline TDS phenotypes also including reduced AGD [124]. This correlation was observed for *ESR1* and *ATF3* genes, whereas no clear association was found between reduced AGD and *AR* gene polymorphisms [124]. Eisenberg et al. further studies this topic, evaluating the association between AGD and CAG repeat length in AR. They noticed that men with a shorter AGD had a higher number of CAG repeats. However, a linear correlation was not established [13]. This finding was in line with the literature considering the indirect association between CAG repeat length and androgenic effects of the interaction between testosterone and its receptor [55,125].

Moreover, not only the length of CAG polymorphism, but also novel SNPs seem to be involved in fertility problems [126]. Genetic alterations of *INSL3* and its receptor *RXFP2* were correlated not only with cryptorchidism but also with azoospermia and crypto-/azoospermia [127,128,129]. In detail, biallelic *RXFP2* variants were associated with histological findings of spermatogenic arrest at the spermatid stage [127]. Evidence in humans regarding the association between AGD and other genes involved in TDS (e.g., *INSL3* or *NR5A1* [79]) are lacking. However, animal models highlighted AGD reduction in association with *NR5A1* mutations [122].

Mutations of *NR5A1* were also found to be involved in male infertility [130]. SNPs seem to play a role in specific populations; however, this finding must be confirmed in different ethnicities [131]. *FGF* family genes were also associated with male infertility [132] and an interaction with *CFTR* in patients affected by azoospermia and cryptorchidism was proposed [133]. Many other genes involved in cryptorchidism, hypospadias, testicular cancer, or reduced AGD, like *ATF3* [134], *DHH* [135], *DMRT1* [136], *EPHX1* [137], *GATA4* [138], *KIT* [139,140], *PIWIL1* [141], *SOX9* [142,143], *WNT* [144], and *WT1* [145,146], were associated with a lower level of evidence with various degrees of semen alterations and male fertility problems.

## 4. Maternal and Fetal Factors Associated with TDS

The association between TDS and environmental factors is well known [9] and many studies evaluated the role of maternal-fetal characteristics, also including exposure to toxins and drugs during pregnancy, in the pathogenesis of the disease.

### 4.1. Characteristics of Pregnancy

Many studies evaluated various maternal and perinatal factors in association with TDS. Zakaria et al. found five conditions associated with higher risk of cryptorchidism, namely, a gestational age of 37 weeks or less, birth weight ≤ 2.75 kg, cesarean delivery, steroid therapy, and twin pregnancy, but the only independent factor predicting cryptorchidism was low birth weight [147]. Maternal age appears to correlate with the risk of cryptorchidism, even though in the study by Stelios Mavrogenis et al. this trend did not reach statistical significance. In the same study, endometriosis in mothers was a risk factor for cryptorchidism in children [148]. In a 1997 study, maternal age over 30 years was associated with increased risk of cryptorchidism (odds ratio 1.9) and seminoma (odds ratio 2.0), especially in the first child. Birth weight, whether excessively low or high, appeared to be associated with increased risk of testicular cancer, particularly in cases of weight < 2.500 kg with an odds ratio of 2.6 [149]. A large Norwegian study in 2004 highlighted the association between hypospadias and several conditions like low parity, hypertension, bleeding during index pregnancy, pre-eclampsia, retained placenta, cesarean section, low birth weight, small for gestational age, presence of inguinal hernia, and prevalence among relatives of hypospadias cases [150]. Another important study involving more than 42 thousand subjects revealed an association between hypospadias and hypertensive disorders of pregnancy (HDP), primiparity, multiple births, hyperthyroidism, preterm delivery, low birth weight (LBW), and small for gestational age (SGA); after correction for potential confounders, some factors remained significant such as HDP, multiple births, and hyperthyroidism [151].

There is evidence that risk of testicular cancer changes in relation to parity. In fact, according to Westergaard et al., there is a reduction in risk of testicular cancer from the second child onwards, which is progressive but only significant when comparing the first child with subsequent children [152]. Similarly, in a large Swedish study, cryptorchidism was associated with being first born [153]. The association between testicular cancer and parity was also confirmed in a 2009 meta-analysis. There was a reduced risk of testicular cancer with increasing parity, with an odds ratio of 0.80 in fourth-born children compared to first-born children. The same article ruled out an association with maternal age, maternal nausea, maternal hypertension, maternal bleeding, pre-eclampsia, breech delivery, and cesarean section [154].

These data are interesting considering that increased parity is usually associated with increased maternal age, which in previous evidence seemed to be associated with higher risk of cryptorchidism and seminoma [149].

Studies on hyperemesis gravidarum were conducted by various authors and were subsequently summarized in the meta-analysis by Nijsten et al., which showed that hyperemesis gravidarum seemed to be associated with risk of various cancers in children, including testicular cancer, with an OR of 1.60 and a 95% confidence interval between 1.07 and 2.39 [155]. The pathophysiological mechanism is unknown, but a correlation with hyperestrogenism has been suggested.

### 4.2. Birth Weight

Birth weight is one of the most extensively studied aspect in relation to TDS, and there are various publications linking low birth weight with cryptorchidism [147,148], hypospadias [150], and testicular cancer [149], as partially mentioned above. In this regard, a retrospective case–control study involving more than 42,000 newborns was published in 2022, showing an association between low birth weight and small for gestational age newborns and hypospadias [151]. Similarly, in 2007, a systematic review found an association between low birth weight and cryptorchidism [156]. However, it is difficult to understand whether low birth weight is independently related to cryptorchidism or if there are common intrauterine factors that determine both conditions, as pointed out by Holmboe et al. [64].

### 4.3. Systemic Arterial Hypertension and Diabetes Mellitus

The literature also focused on maternal hypertension in relation to TDS. A systematic review with meta-analysis in 2009 did not find associations between hypertension or pre-eclampsia and testicular cancer in sons [154], while other papers showed a link between HDP and hypospadias as mentioned above [150,151].

Another interesting aspect is the role of pregestational and gestational diabetes in mothers of children affected by TDS. Pregestational diabetes appears to be associated with increased risk of congenital anomalies, probably due to prolonged exposure to hyperglycemia. It leads to increased reactive oxygen species (ROS) production and oxidative stress, compromising organogenesis and interfering with normal embryonic development [157]. The development of the male reproductive system is a process driven by androgenic production by the fetus in response to the placenta’s chorionic gonadotrophin (hCG) production. In diabetic women, placental abnormalities may develop, compromising its normal secretion and thus the masculinization process [158]. As observed by Arendt et al., pregestational diabetes, particularly type 1 diabetes mellitus, is associated with increased risk of hypospadias and cryptorchidism [158].

In 2006, Virtanen H.E. et al. showed that gestational diabetes was more common in mothers of patients with cryptorchidism, even in cases of mild gestational diabetes treated with diet alone. However, the authors themselves pointed out that the criteria used for the diagnosis of gestational diabetes were not described [159]. A 2020 study conducted in the US on data concerning more than twenty-nine million newborns showed an association between diabetes during pregnancy and hypospadias, with an adjusted risk ratio of 1.88 (95% CI 1.67–2.12) for pre-pregnancy diabetes and 1.29 (95% CI 1.21–1.36) for gestational diabetes [160]. Less-impressive data came from a systematic review with meta-analysis conducted in 2015 on thirteen studies including 15,373 cases, which highlighted that gestational diabetes was only marginally associated with increased risk of cryptorchidism with a trend towards statistical significance that was not, however, achieved (OR = 1.21, 95% CI: 1.00–1.46; *p* = 0.06) [161].

### 4.4. Maternal Weight and Physical Activity

Maternal weight was also studied in relation to TDS. Studies focused not only on pre-pregnancy weight but also on weight gain during pregnancy. Overweight and obesity can be associated with increased estrogens levels, as well as poorer nutritional status and glycemic control [162]. These alterations could interfere with genital development and testicular descent [35]. For example, sons of overweight or obese women had higher risk of developing hypospadias [163,164,165]. Similarly, a Swedish study on 1,055,705 infants highlighted an association between maternal overweight or obesity and cryptorchidism or hypospadias [35]. An interesting Chinese study analyzed the AGD of 556 male children, revealing that it was shorter in children whose mothers had gained more weight during pregnancy. In fact, AGD in male newborns decreased about 2.27 mm in the excessive-gestational-weight-gain group compared to the standard-gestational-weight-gain group [166].

Nutrition may also play a role in the development of defects in the male genital tract. A single study suggested that a vegetarian diet during pregnancy could be associated with increased risk of hypospadias, probably due to excessive intake of phytoestrogens [167]. However, this finding was not confirmed in more recent evidence [168].

Evenson et al. observed that maternal physical activity and reduced sedentary behavior were associated with lower risk of hypospadias [169], but no further studies are currently available on this topic. The result was probably related to better weight control and reduced risk of type 2 diabetes.

### 4.5. Smoking Habit

Various studies evaluated exposure to smoke during pregnancy and hypospadias in newborns. A recent meta-analysis of observational studies concluded that maternal active smoking was significantly associated with risk of hypospadias, whereas this was not true for paternal smoking nor maternal passive smoking [170]. Similarly, a large study based on nationwide birth certificate data revealed an increased risk of several malformations, including hypospadias, in children born from mothers who smoked during pregnancy [34]. The association between smoking during pregnancy and hypospadias has also been confirmed by large reviews and meta-analyses [161,171,172] and this phenomenon seems dose-dependent, in terms of number of cigarettes per day and nicotine content [173]. However, some studies showed opposite results, highlighting a reduced risk of hypospadias in children whose mothers smoked during pregnancy. In particular, the study by Daniel Lindbo et al. showed that this kind of exposure was associated with a lower risk of hypospadias with an adjusted hazard ratio of 0.82 and a 95% confidence interval between 0.58 and 1.18 [174]. Previously, a systematic review had found similar results, suggesting a reduction in risk of hypospadias considering 15 studies with 12,047 malformed cases and 1.5 million controls [171]. Moreover, there is evidence supporting a neutral effect of cigarette smoking during pregnancy on risk of hypospadias [175]. Finally, another study found that passive maternal exposure to tobacco smoke during pregnancy was associated with an 88% increased risk of hypospadias, but this result was not confirmed for active maternal smoking in the first trimester [176]. In conclusion, the variability of population examined and different study designs probably led to the different results previously described. For that reason, it is not possible to draw any definite conclusions regarding the impact of maternal smoking on hypospadias considering the available evidence.

Otherwise, the relationship between maternal smoking during pregnancy and cryptorchidism seems to be more consistent. In the study by Zakaria et al., maternal exposure to passive smoking was more common in patients with cryptorchidism than in controls, although statistical significance was not achieved [147]. In a Danish cohort study, authors demonstrated that sons of women who smoked 10–14 cigarettes/day had the highest hazard ratio for cryptorchidism (1.37; 95% CI: 1.06–1.76) and that smoking cessation was associated with a slightly higher HR of cryptorchidism compared to non-smokers [177]. The study by Lindbo et al. also demonstrated a dose-dependent relationship [174].

Finally, testicular cancer did not seem to be associated with smoking during pregnancy in a 2009 study, even with dividing the population into passive or active smokers or considering the number of cigarettes smoked daily [33,178].

### 4.6. Alcohol and Caffeine

The role of alcohol consumption in risk of TDS is under debate. According to Mongraw-Chaffin et al., mothers of sons with testicular cancer were more likely to drink alcohol than controls [33]. Other more recent studies did not find an association between maternal alcohol consumption and TDS components, in particular cryptorchidism [161,177]. Despite mixed results, it is interesting to consider that the first study described [33] had the advantage of being based on older data. At the time of data collection, alcohol consumption during pregnancy was not as stigmatized as today due to less knowledge of the negative effects of alcohol on the fetus at the time. For this reason, medical histories in this regard could be reasonably more reliable. However, the more-recent evidence has the strength of analyzing a larger population.

Regarding coffee consumption during pregnancy, no associations with increased risk of cryptorchidism were found [177]. It was noted that mothers of patients affected by testicular cancer were less likely to drink coffee than controls [33].

### 4.7. Assisted Reproductive Technology

There are several studies in the literature, which can be summarized by the meta-analysis by Zhang et al. [179], which showed that children born with Assisted Reproductive Technology (ART) had a 1.87 times higher risk of hypospadias than naturally conceived male offspring. The meta-analysis was affected by moderate heterogeneity. However, the risk was confirmed after exclusion of studies mainly implicated in the heterogeneity. The same meta-analysis showed no differences in risk of hypospadias or cryptorchidism when comparing In Vitro Fertilization (IVF) and Intracytoplasmic Sperm Injection (ICSI) techniques [179]. Other studies found ICSI to present higher risk of hypospadias and cryptorchidism than IVF. In any case, it has been hypothesized that the association between ART, cryptorchidism, and hypospadias is because newborns conceived through ART techniques are more frequently affected by low birth weight and prematurity [180].

### 4.8. Drugs

The Dorte Vesterholm Lind group evaluated the relationship between exposure to Non-Steroidal Anti-inflammatory Drugs (NSAIDs) and/or paracetamol (acetaminophen) and AGD. In the study, combined exposure to NSAIDs and paracetamol was associated with a statistically significant reduced AGD compared to non-exposed subjects (mean: 32.3 mm versus 36.2 mm, *p* = 0.03). The low sample size must be considered among the limitations of the study. This result was not confirmed considering exposure to paracetamol alone [181]. In fact, Navarro-Lafuente et al. did not find an association between exposure to paracetamol alone and AGD [182].

Allopurinol is not recommended during pregnancy, mainly due to a lack of data. In the prospective case series study conducted by Hoeltzenbein et al. on 31 unborn babies exposed to the drug in the first trimester, the overall rate of major malformations (3.7%) and spontaneous abortions (cumulative incidence 11%, 95% CI 3–40) were both within the normal range [183]. However, one fetus was described as having severe malformations (including microphthalmia, cleft lip and palate, renal hypoplasia, low-set ears, hearing deficit, bilateral cryptorchidism, and micropenis), which were like those described by Kozenko et al., leading to the suspicion that allopurinol may cause malformations related to TDS [184].

Antiepileptic drugs are statistically associated with fetal malformations when used during pregnancy. According to the meta-analysis by Veroniki et al., gabapentin (OR, 16.54; 95% CI, 2.50–121.70), primidone (OR, 5.92; 95% CI, 1.01–23.77), and valproic acid (OR, 2.58; 95% CI, 1.24–5.76) in monotherapy were associated with an increased risk of hypospadias, while the drugs least associated with malformations were levetiracetam and lamotrigine [185].

It is well known that the use of beta-blockers during pregnancy is associated with an increased risk of malformations. However, this increased risk does not appear applicable to hypospadias, as shown by two different meta-analyses of observational studies carried out several years apart [186,187].

Regarding exposure to macrolides, a systematic review with meta-analysis was published in 2023, which ruled out any association between hypospadias and use of macrolides during pregnancy. Considering two studies that specifically evaluated the use of erythromycin in the first trimester, the drug was significantly associated with a reduction in risk of hypospadias (OR 0.38; 0.18, 0.81), but given the small number of cases evaluated, the authors hypothesized that this was a chance finding [188].

### 4.9. Socioeconomic Status

Socioeconomic factors also appear to be involved in TDS and these aspects were already studied in the early 1990s. In 1996, a large Danish study was conducted, highlighting a weak association between testicular cancer and higher socioeconomic status, while cryptorchidism was associated with lower socioeconomic status. However, even at that time, the authors hypothesized that the study of socioeconomic factors is difficult for various reasons, including historical changes that, for example, may render past results no longer valid [189]. However, similar results were found in another study on a Hungarian population, revealing a lower socioeconomic status of mothers of cryptorchid newborns [148]. In any case, data on this topic is insufficient and no conclusions can be drawn.

## 5. Paternal Factors Associated with TDS

Studies mainly focused on maternal risk factors of TDS, considering the importance of gestational period on fetal organogenesis. However, there is also evidence regarding the impact of paternal risk factors. In 2017, the analysis of a population-based cohort of 1,056,058 males found that younger paternal age at birth was associated with increased risk of TGCT, especially seminoma, in sons. The association was confirmed even after adjustment for year of birth, years of education, height, history of cryptorchidism, and maternal age at birth [190]. Moreover, a direct correlation between older age of fathers and impaired sperm quality of children was observed [191]. Then, a case–control study found that paternal smoking (OR 2.0; 95% CI: 1.33–2.99), history of urological diseases (OR 2.31; 95% CI: 1.15–4.90), and occupational exposure to EDC (OR 3.90; 95% CI: 2.41–6.48) were risk factors for hypospadias and cryptorchidism in sons, whereas higher paternal educational level seemed to be a protective factor (OR 0.63; 95% CI: 0.42–0.93) [192]. An interesting study on 376,362 male births also highlighted the impact of paternal metabolic health on the risk of genital malformations in children. Indeed, sons of men affected by two or more components of metabolic syndrome were more likely to be diagnosed with hypospadias (OR 1.27 95% CI: 1.10–1.47) [193]. Considering the available evidence, similar risk factors on both maternal and paternal sides seem to impact on the risk of TDS in sons.

## 6. Endocrine-Disrupting Chemicals

Endocrine-disrupting chemicals (EDCs) are exogenous compounds that interfere with the endocrine system by mimicking, antagonizing, or altering synthesis, metabolism, and signaling of hormones. In the 1990s, researchers first noticed a connection between growing pollution and progressive decline in male fertility and rising incidence of male genitalia abnormalities, including testicular tumors (a cluster later defined as TDS). Evidence then confirmed this connection [36,194]. Different definitions of EDCs have been proposed since then, and currently the Endocrine Society describes EDCs as exogenous substances, or mixtures of chemicals, that interfere with any aspect of hormone function [195]. From a regulatory perspective, these substances are sorted by European Society into three groups—“Identified”, “Under Evaluation”, and “Considered by National Authority as EDCs”. Among more than 800 chemicals that might have potential endocrine action [23], researchers mainly focused on phthalates, phytoestrogens, pesticides, bisphenols, per- and polyfluoroalkyl substances (PFAS), and polychlorinated biphenyls (PCBs) because they are the more commonly used and more relevant in clinical practice.

### 6.1. Mechanism of Action

The main mechanism linking EDCs to TDS is the ability of these compounds to interfere with androgen and estrogen signaling. Indeed, they can bind hormonal receptors acting as agonists or antagonists. For example, Bisphenol A (BPA) can bind the main nuclear estrogen receptor (ER), exerting an agonistic effect even at low concentrations, possibly altering the physiologic hormonal balance. BPA can also bind other receptors of the estrogen pathway like estrogen-related receptor γ [196], G protein-coupled receptors, and GPR3 0 [197]. In this way, BPA could interfere with germ cell proliferation and differentiation and may also induce pro-apoptotic effects on spermatocytes [198]. Finally, BPA could potentially block or interfere with AR, the progesterone receptor, and their ligands [199]. Interestingly, several studies showed that BPA can induce changes in DNA methylation and consequently lead to epigenetic changes transmissible from exposed mothers to their offspring; however, further studies are needed to translate this mechanism into correlation with phenotypic outcomes [200]. Among pesticides, although it has been banned for many years, Dichlorodiphenyltrichloroethane (DDT), an organochlorine, represents a clear example of the harmful potential of EDCs. It is capable of binding with high affinity both the nuclear ER [201] and G-protein-coupled receptors [202], and this could translate into effects on spermatogenesis and gonadal development. PCBs are also compounds with estrogenic effects [203] and can cause alterations in the development of reproductive organs and impair the hypothalamus–pituitary–gonadal axis. Epigenetic modifications transmitted between generations were documented for PCBs, although a direct correlation between these modifications and components of TDS is still lacking [204]. Phthalates can directly affect steroid hormone production and increase ROS-derived damage both directly and indirectly, altering antioxidant enzyme activity [205]. Some compounds can act as agonists and others as antagonists on ER, but at AR level they are almost exclusively antagonists [206]. Animal evidence also showed a possible role in the apoptosis of testicular cells [207]. Particular concern regards the epigenetic alterations potentially associated with all the compounds mentioned above. They represent the mechanism by which EDCs modify gene expression without altering the DNA sequence. The main epigenetic changes highlighted so far are aberrant methylations, histone modifications, and alterations in the microRNA profile [208]. In summary, the main mechanisms by which EDCs exert their effects, potentially compromising male reproductive health, include receptor binding, oxidative stress, steroidogenic disruption, and epigenetic modifications. In addition, emerging evidence shows that reproductive adverse effects of EDCs may not be limited to the exposed generation of individuals but could be inherited by future generations, possibly impairing reproductive health and susceptibility to reproductive disorders [209].

### 6.2. Evidence Regarding TDS

In all experimental models, phthalates exposure, particularly di-2-ethylhexyl phthalate/dibutyl phthalate (DEHP/DBP), can reduce testosterone and INSL3 levels as well as AGD, and is associated with an elevated risk of cryptorchidism, with the risk being higher for mixtures of compounds [12]. However, in humans, maternal gestational urinary bisphenols and phthalates concentrations did not seem to be associated with increased risk of cryptorchidism in offspring [210]. BPA seems to reduce testosterone levels in experimental studies, but the reduction is relevant only in the absence of hormones stimulating gonadal function; in the presence of hCG or LH, the differences are irrelevant, suggesting a potential modulatory effect of the placenta on this outcome [211]. Evidence from epidemiological studies is scarce and association was found only for high levels of exposure. More recent and longitudinal studies did not find an association at low exposure levels, and results are very heterogeneous and difficult to interpret [212]. The most robust evidence exists for pesticides, which showed an augmented risk of cryptorchidism, but there is a lot of heterogeneity in the methods of studies carried out: this includes different matrices or indirect evaluation of exposures, small sample size, and diverse geographic origins of populations. Parabens, like BPA, were linked to cryptorchidism but only for high levels of exposure [64]. A clear association with PFAS is yet to be found, with only a few studies showing a possible link [43]. Similarly, only limited evidence exists for PCBs, with most studies showing no association [64]. In a recent retrospective cohort study on a group of 253 XY individuals who were exposed in utero to diethylstilbestrol (DES), a drug utilized for miscarriage and now ubiquitously banned, Gaspari et al. [213] described two cases of cryptorchidism. Interestingly, those individuals (and two other subjects) reported a female transgender identity later in life. A relationship between hormonal exposure in utero and gender identity has been suggested, and it is known that genes and gonadal hormones can have an effect on psychosexual development, shaping the sexual dimorphism of the brain and contributing to the formation of gender identity [214]. Accordingly, it is possible that endocrine disruptors may also play a role in this aspect, and the hypothesis is fascinating, but the current evidence is too scarce to be able to express a definitive opinion.

Evidence on hypospadias led to ambiguous results, and possible biases should be considered. In fact, prevalence and incidence of this condition are lower, and mild cases may go undiagnosed in early childhood. Initial evidence that altered estrogen balance could lead to hypospadias was derived from studies on DES, particularly if ingestion occurred during the MPW [215]. Regarding pesticides, results are mixed, with potential risk emerging from maternal exposure to elevated concentrations of dichlorodiphenyltrichloroethane (DDT) and its metabolites [216]. Atrazine showed in mice the potential to reduce and shorten AGD; however, human studies showed a non-linear and inconsistent association for hypospadias [217,218]. The mother’s exposure to insect repellents during the first trimester may also be associated with an increased risk for this outcome [219]. For PCBs, most evidence is against their role in the pathogenesis of hypospadias; only a single study found a weak association, but it was non-linear [220]. A study evaluating the combination of a fungicide and PCB showed a link with the fungicide only, with no correlation to PCB [221]. Evidence showed a correlation between hypospadias and phthalate concentrations in blood of children affected by this condition, but not with maternal phthalate levels [222]. Another study evaluating blood samples of children at birth showed an inverse correlation between low-weight phthalates and hypospadias [223]. Evaluation of amniotic fluid in the second trimester (an indication linked to maternal age) showed a weak association for DEHP and no statistically significant association for Diisononylphthalate (DINP) [224]. Even though most human studies are inconclusive, there is biological plausibility (mostly through antiandrogenic effects and more generally through alteration of the androgen–estrogen balance) and epidemiological evidence of pollutant impact from industrialized regions on both cryptorchidism and hypospadias [64,225].

Increasing incidence of testicular cancer has been observed in epidemiological studies that cannot be justified only by genetics, highlighting a possible role of environmental factors [226]. Evidence showed that TGCC has a fetal origin, with pre-cancerous lesions forming during gestation, and there is also a significant association between TGCC and both cryptorchidism and hypospadias, suggesting a possible common origin [227,228]. Studies in humans regarding the impact of EDCs on the pathogenesis of testicular cancer are mainly retrospective. Most of them evaluated persistent EDCs; only a few were conducted on more rapidly metabolized ones (i.e., phthalates and BPA) [229]. With regard to different classes of xenobiotics, pesticides, particularly organochlorines, present some evidence of association with testicular cancer; however, other studies did not find significant correlations [230,231]. A positive association exists for organ halogens, especially for prenatal/maternal exposure; however, some studies controversially showed that postnatal exposure decreases the overall risk of testicular tumors. These results are difficult to interpret, and no clear biological explanation is currently published [232]. Both organochlorines and organ halogens showed an associated increased risk of nonseminoma compared to seminoma, possibly reflecting differences in age at onset [232]. PCBs presented mixed evidence, with some studies showing associations between certain PCB congeners and seminoma patients, while others showed a significantly lower risk compared to controls [233,234]. Limited and mixed evidence of association exists for PFAS, phthalates, and BPA [229,235]. Overall, the data showed an increased risk for fetal windows of exposure, supporting the developmental origin of TGCC; in contrast, postnatal exposures do not appear linked to increased risk [236]. As for cryptorchidism and hypospadias, the plausibility of a link between TGCC and EDCs emerges from studies in the current literature, but methodological problems limit the quality of available evidence, particularly in humans.

The characteristics of the main ECDs are summarized in Table 3.

## 7. Shared Risk Factors Among TDS Components

To better understand possible the etiopathogenetic mechanisms of TDS, it is useful to focus on shared risk factors among its components. The most common genetic risk factors regard *AR* [76,115], *INSL3* [52,128], *RXFP2* [127], and *NR5A1* [80,130]. Exposure to EDC seems to be a common risk factor for all TDS components [64,225,236], strengthening the hypothesis of its main role in the pathogenesis of the disease. Among maternal and fetal risk factors, low birth weight is a shared common risk for cryptorchidism, hypospadias, and testicular cancer [149,176]. It is also possible that the association found with ART is related to increased risk of low birth weight in this condition [180]. The role of maternal and paternal components of metabolic syndrome is interesting considering that they are of common clinical practice management. In fact, overweight and diabetes, mainly gestational diabetes, seem to be associated with increased risk of hypospadias and cryptorchidism [35,193].

## 8. Possible Mechanistic Models of Synergic Effect of Environmental Factors and Genetic Background on Pathogenesis of TDS

Experimental models to demonstrate the interaction of genetic variants with environmental disruptors were built (Figure 2). Many laboratories used cultures of human testis cell to evaluate the effects of EDCs and drugs on fetal testis. Phthalates exposure led to increased apoptosis of germ cells and bisphenols to reduced testosterone production and INSL3 transcription. Interestingly, paracetamol seemed also to reduce INSL3 production and gonocyte proliferation [12]. Studies using xenograft models of human fetal testis tissue were also carried out. Contrasting results were found, considering that exposure to DBP did not reduce the expression of genes involved in testosterone biosynthesis [242]. Instead, paracetamol exposure reduced testosterone production and lowered seminal vesicle weight in second-semester human fetal testis [12,243]. Mechanistic studies directly evaluating the effects of environmental factors on the presence of genetic variants in TDS models are missing. However, data deriving from mice affected by DSD, a conceptually similar condition, can be analyzed. Xie et al. found that the effects of exposure to different doses of DEHP were different in the presence of a specific genetic variant of the luteinizing hormone/choriogonadotropin receptor (LhcgrW495X/+). Indeed, low-dose DEHP did not exert significant effects on wild-type mouse offspring but caused DSD in LhcgrW495X/+ mice by suppressing the expression of genes involved in steroidogenesis. Moreover, even if high-dose DEHP caused DSD in both genotypes, LhcgrW495X/+ mice had more severe phenotypes [244]. It is possible that a similar mechanistic model could also be applicable to TDS considering the similar etiopathogenesis. Unfortunately, building a mechanistic model evaluating the effects of many maternal and fetal risks previously listed is quite difficult, even if it would surely add important information.

## 9. Conclusions

TDS is a complex syndrome. Concerning genetics, many genes correlated to various components of the syndrome, and others will probably be discovered in following years. Environmental factors and maternal habits are also associated with TDS and represent modifiable factors that should be identified and addressed. Prospective, multicentric, well-designed studies could add important information to this topic, leading to a significant impact on clinical practice as well as a better understanding of the pathogenesis of TDS.

## Figures and Tables

**Figure 1 genes-17-00040-f001:**
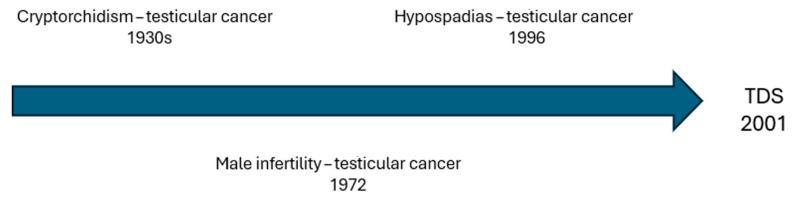
Temporal trend of associations between components of testicular dysgenesis syndrome (TDS).

**Figure 2 genes-17-00040-f002:**
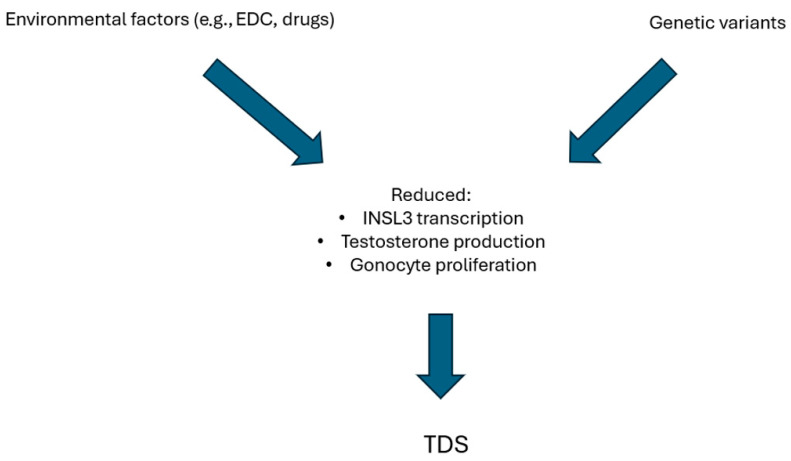
Hypothesis of mechanistic models of synergic effect of environmental factors and genetic background on pathogenesis of TDS.

**Table 1 genes-17-00040-t001:** Summary of genes involved in cryptorchidism in TDS. INSL3 = Insulin-like factor 3, AR = androgen receptor, DSD: disorders of sex development, KCTD13 = potassium-channel-tetramerization-domain-containing-13, DMRT1 = Doublesex and Mab-3 Related Transcription Factor 1, SNP = Single Nucleotide Polimorphism.

Gene	Function/Pathway	Documented Role	Evidence/Notes Regarding Cryptorchidism
*INSL3*	Fetal hormone involved in transabdominal testicular descent	Rare mutations associated with bilateral cryptorchidism; essential in mouse models [49,51]	Recessive variants can cause isolated cryptorchidism
*RXFP2*	Receptor of INSL3	Mutations linked to persistent or bilateral cryptorchidism; limited role in sporadic cases [52]	Some variants confirmed in families with autosomal recessive inheritance
*AR*	Androgen receptor	CAG/GGN repeat polymorphisms linked to reduced androgen sensitivity [55]	Association with cryptorchidism remains poorly replicable
*NR5A1*	SF-1, regulator of steroidogenesis	Mutations linked to DSD, infertility, and variable phenotypes; occasionally present in cryptorchidism [53,54]	Responsible for ~10–15% of DSD cases; cryptorchidism often part of broader phenotype
*WT1, SOX9, GATA4, DHH*	Testicular development and gonadal differentiation	Variants associated with gonadal dysgenesis and testicular dysfunction [53,54]	Indirect or secondary role in cryptorchidism
*KCTD13*	Unknown role in genitourinary tract. Loss of KCTD13 in germ lines was associated with decreased intracellular AR levels.	Prevalence of variants significantly higher in patients with genitourinary abnormalities in comparison to control [56]	Haploinsufficiency and homozygote deletion in mice caused cryptorchidism.
*DMRT1*	Germ cell regulation and Sertoli cell differentiation	Deletions/mutations linked to gonadal dysgenesis and infertility; possible link to testicular cancer [59]	Limited direct evidence for cryptorchidism involvement
*AXIN1*	WNT pathway	SNPs associated with increased risk of cryptorchidism [60]	Evidence derived from small cohorts
*ATRX, PIWIL1, CPEB1, DAZL*	Spermatogenesis regulation and germ cell development	Prioritized from gene network analysis and human/mouse expression data [61,62]	Indirect but potentially relevant contribution

**Table 2 genes-17-00040-t002:** WHO classification of testicular cancer. GCT = germ cells tumor, GCNIS = germ cell neoplasia in situ, NET = neuroendocrine tumor.

**Germ Cell Tumors**	
Derived from GCNIS	Unrelated to GCNIS
Seminoma	Spermatocytic tumor Yolk sac tumor (prepubertal-type) Teratoma (prepubertal-type) Testicular NET
Non seminomatous GCTs Embryonal carcinoma Yolk sac tumor (postpubertal-type) Teratoma (postpubertal-type) Trophoblatic tumors ○ Choriocarcinoma ○ Placental site trophoblastic tumor ○ Epithelioid trophoblastic tumor Mixed GCTs	
**Sex Cord Stromal Tumors**	
Leydig cell tumor	
Sertoli cell tumor	
Granulosa cell tumor	
The fibroma thecoma family of tumors	
Mixed and other sex cord stromal tumors	

**Table 3 genes-17-00040-t003:** Main features of the most common EDCs.

Class	Main Uses	Current Regulatory Status (Banned/Restricted)	Metabolism and Exposure
Phthalates	Plasticizers in packaging, medical devices, toys, and personal care products.	Some (e.g., diethylhexyl phthalate, dibutyl phthalate) restricted or banned in the EU and USA due to reproductive toxicity.	Rapid metabolism: hydrolyzed to monoesters and conjugated; excreted in urine. Exposure occurs via ingestion, inhalation, and dermal contact [237].
Phenols (e.g., Bisphenol A, Bisphenol S)	Used in the production of polycarbonate plastics, epoxy resins, food packaging, and thermal paper.	Bisphenol A banned in baby bottles and food-contact materials in the EU and several countries; BPS and BPF increasingly monitored as substitutes.	Relatively rapid metabolism through conjugation (glucuronidation/sulfation); exposure mainly via ingestion, inhalation of dust, and dermal contact [238]
Pesticides	Insecticides, herbicides, and fungicides used in agriculture and public health.	Many banned or restricted in the EU and North America; some persistent organic pollutants still detected globally.	Variable: organophosphates are metabolized quickly, while organochlorines are persistent and bioaccumulative. Exposure through food, air, and occupational contact [239]
Polychlorinated Biphenyls (PCBs)	Formerly used as dielectric fluids in transformers, capacitors, and lubricants.	Banned worldwide since the 1970s–1980s under the Stockholm Convention, but residues persist in the environment.	Very slow metabolism; lipophilic and bioaccumulative in adipose tissue; exposure mainly via diet (fish, meat, dairy) [240].
Per- and polyfluoroalkyl substances PFAS	Water- and oil-repellent coatings, firefighting foams, textiles	Perfluorooctanoic acid (PFOA) and Perfluorooctanesulfonic acid (PFOS) restricted or banned in EU and USA for certain industrial uses	Highly persistent, bioaccumulative; main exposure via food and water [241]

## Data Availability

No new data were created or analyzed in this study. Data sharing is not applicable to this article.

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
