# Peer review of "Genetic, Epigenetic, and Non-Genetic Factors in Testicular Dysgenesis Syndrome: A Narrative Review"

_genes, 2025, doi:10.3390/genes17010040_

Round 1
Reviewer 1 Report
Comments and Suggestions for Authors
Comment 1
The authors aim to critically evaluate the genetic and non-genetic factors involved in the pathogenesis of TDS and to clarify the possible approaches to mitigate its increasing incidence. The effort to synthesize a broad range of findings based on extensive literature is commendable.
However, the manuscript would benefit from a clear description of the literature-search strategy, which is essential for transparency in any narrative review. The authors are encouraged to describe how the references were selected and which databases or keywords were used. Reference to the Scale for the Assessment of Narrative Review Articles (SANRA) could be useful in strengthening methodological rigor.
Comment 2
Although cryptorchidism, hypospadias, and testicular cancer are long-recognized entities, the concept of testicular dysgenesis syndrome (TDS) integrating these conditions has been proposed only in recent decades. A brief historical explanation of how the TDS concept emerged would help readers understand the background and rationale for this integrative framework.
Comment 3
In Section 2 “Genetic and epigenetic causes of TDS,” the authors discuss cryptorchidism, hypospadias, and testicular cancer in detail. Since male infertility represents another major component of TDS, the review would be more balanced if infertility were also discussed at a comparable level, including its genetic and epigenetic aspects.
Comment 4
In Section 2 “Genetic and epigenetic causes of TDS,” shorter anogenital distance (AGD) is treated alongside cryptorchidism, hypospadias, and testicular cancer. Because AGD is not generally regarded as a major component of TDS but rather a related or secondary indicator, readers may find this structure confusing. The authors might consider creating a separate subsection dedicated to minor or associated features of TDS, where AGD could be discussed as an illustrative example. This restructuring would improve logical clarity and reader comprehension.
Comment 5
To further clarify the distinction between TDS and its individual disease components, it would be helpful to evaluate both the disease-specific and the shared genetic and non-genetic factors for cryptorchidism, hypospadias, and testicular cancer. Such a comparative summary could highlight overlapping mechanisms as well as features unique to each condition, enhancing readers’ understanding of how TDS functions as an overarching pathophysiological concept.
Comment 6
The manuscript states that the incidence of TDS cases has been increasing in developed countries. However, it remains unclear whether this increase is merely a general trend or whether there is any causal relationship supported by epidemiological or experimental evidence. If data from epidemiological studies or animal models exist to substantiate a causal link, it would be valuable for the authors to review such literature and provide their interpretation.
Comment 7
Although environmental pollution is generally considered more severe in developing countries, the increasing incidence of TDS in developed nations appears, at least superficially, contradictory. If there are studies that help reconcile this discrepancy—such as those discussing differences in exposure types, reporting accuracy, or healthcare access—the authors should introduce and discuss them to enhance readers’ understanding.
Comment 8
A recent global meta-analysis (Reference 141, PMID: 36377604) reported a declining trend in semen quality worldwide, which has attracted significant attention in the scientific community. Because this issue is closely related to male reproductive health, readers will likely find it of great interest. If the decline in semen quality can be conceptually or mechanistically linked to TDS, the authors are encouraged to discuss this relationship with reference to the relevant literature and provide their own perspective.
Comment 9
Reports have shown a declining trend in serum testosterone levels among young men (PMID: 32081788) and an increasing prevalence of health complaints associated with low testosterone (PMID: 40077734). These findings are also of great public and academic interest. If these trends share a biological basis with the rising incidence of TDS, the authors should discuss the potential connections and interpret them in light of existing evidence.
Comment 10
Several studies have reported that the age at onset of testicular cancer has been increasing in developed countries (PMID: 32627833, PMID: 23395239). Given that such demographic shifts could influence treatment strategies, this topic would likely engage readers. If these temporal changes are potentially related to TDS, the authors should provide a literature-based discussion and offer their interpretation.
Comment 11
The manuscript divides testicular tumors into germ cell tumors and sex cord stromal tumors. Although sex cord stromal tumors are much less common and the literature on them is limited, they may share biological mechanisms with TDS. It would strengthen the review if the authors could check whether any references discuss this possibility and briefly summarize their opinion on the potential overlap.
Comment 12
The authors cite Reference 60 and discuss Disorders of Sex Development (DSD), congenital conditions involving atypical development of chromosomal, gonadal, or anatomical sex. Because DSDs partially overlap conceptually with hypospadias and cryptorchidism yet differ in other respects, a short literature-based explanation distinguishing DSDs closely related to TDS from those less related would help readers grasp the nuanced relationship between these entities.
Comment 13
Recent studies have gradually accumulated evidence suggesting that biological, anatomical, genetic, and environmental factors may contribute to gender identity. Particularly noteworthy are reports of transgender cases associated with cryptorchidism (PMID: 38249107). Although this is a highly sensitive topic and must be handled with care, if epidemiological or mechanistic evidence suggests any overlap between factors contributing to transgender identity and those underlying TDS, it would be worthwhile to mention this cautiously with appropriate references and interpretation.
Comment 14
Section 3, Maternal and fetal factors associated with TDS, provides a comprehensive discussion of maternal risk factors, but there is no mention of paternal risks. This omission could leave readers with an impression of imbalance. If relevant studies exist addressing paternal factors—such as the effects of paternal aging or environmental exposures—the authors should cite and discuss them, presenting their own perspective where possible.
Author Response
Comment 1
The authors aim to critically evaluate the genetic and non-genetic factors involved in the pathogenesis of TDS and to clarify the possible approaches to mitigate its increasing incidence. The effort to synthesize a broad range of findings based on extensive literature is commendable.
However, the manuscript would benefit from a clear description of the literature-search strategy, which is essential for transparency in any narrative review. The authors are encouraged to describe how the references were selected and which databases or keywords were used. Reference to the Scale for the Assessment of Narrative Review Articles (SANRA) could be useful in strengthening methodological rigor.
Reply 1
We are extremely grateful to the reviewer for the time dedicated to evaluating our work and for the valuable advice. We used the SARNA scale to ensure that the research methodology was transparent and consistent. As a result, a section describing the research strategy has been added.
Comment 2
Although cryptorchidism, hypospadias, and testicular cancer are long-recognized entities, the concept of testicular dysgenesis syndrome (TDS) integrating these conditions has been proposed only in recent decades. A brief historical explanation of how the TDS concept emerged would help readers understand the background and rationale for this integrative framework.
Reply 2
We thank the reviewer for the suggestion; we added Figure 1 and a brief history of TDS (lines 39-47).
Comment 3
In Section 2 “Genetic and epigenetic causes of TDS,” the authors discuss cryptorchidism, hypospadias, and testicular cancer in detail. Since male infertility represents another major component of TDS, the review would be more balanced if infertility were also discussed at a comparable level, including its genetic and epigenetic aspects.
Reply 3
Thank you for highlighting the important point, we added a description of genetic and epigenetic aspects of infertility in TDS (Section 3.4 – Lines 324-375)
Comment 4
In Section 2 “Genetic and epigenetic causes of TDS,” shorter anogenital distance (AGD) is treated alongside cryptorchidism, hypospadias, and testicular cancer. Because AGD is not generally regarded as a major component of TDS but rather a related or secondary indicator, readers may find this structure confusing. The authors might consider creating a separate subsection dedicated to minor or associated features of TDS, where AGD could be discussed as an illustrative example. This restructuring would improve logical clarity and reader comprehension.
We thank the reviewer for highlighting this point and helping to clarify the text. We have moved the section on anogenital distance to the section on male infertility, clarifying its role as a “marker” of androgenization rather than a component of TDS.
Comment 5
To further clarify the distinction between TDS and its individual disease components, it would be helpful to evaluate both the disease-specific and the shared genetic and non-genetic factors for cryptorchidism, hypospadias, and testicular cancer. Such a comparative summary could highlight overlapping mechanisms as well as features unique to each condition, enhancing readers’ understanding of how TDS functions as an overarching pathophysiological concept.
Reply 5
We thank the reviewer for the suggestion, we added the section 7 aiming to better summarise shared risk factors among components of TDS (lines 781-792)
Comment 6
The manuscript states that the incidence of TDS cases has been increasing in developed countries. However, it remains unclear whether this increase is merely a general trend or whether there is any causal relationship supported by epidemiological or experimental evidence. If data from epidemiological studies or animal models exist to substantiate a causal link, it would be valuable for the authors to review such literature and provide their interpretation.
Reply 6
We sincerely thank the reviewer for this comment. We have discussed the topic in greater detail in the relevant section.
Comment 7
Although environmental pollution is generally considered more severe in developing countries, the increasing incidence of TDS in developed nations appears, at least superficially, contradictory. If there are studies that help reconcile this discrepancy—such as those discussing differences in exposure types, reporting accuracy, or healthcare access—the authors should introduce and discuss them to enhance readers’ understanding.
Reply 7
We sincerely thank the reviewer for this comment. We have discussed the topic in greater detail in the relevant section.
Comment 8
A recent global meta-analysis (Reference 141, PMID: 36377604) reported a declining trend in semen quality worldwide, which has attracted significant attention in the scientific community. Because this issue is closely related to male reproductive health, readers will likely find it of great interest. If the decline in semen quality can be conceptually or mechanistically linked to TDS, the authors are encouraged to discuss this relationship with reference to the relevant literature and provide their own perspective.
Reply 8
Thank you for the comment, we discussed the interesting topic in the introduction (lines 72-76).
Comment 9
Reports have shown a declining trend in serum testosterone levels among young men (PMID: 32081788) and an increasing prevalence of health complaints associated with low testosterone (PMID: 40077734). These findings are also of great public and academic interest. If these trends share a biological basis with the rising incidence of TDS, the authors should discuss the potential connections and interpret them in light of existing evidence.
Reply 9
Thank you for the comment, we discussed the interesting topic in the introduction (lines 72-76).
Comment 10
Several studies have reported that the age at onset of testicular cancer has been increasing in developed countries (PMID: 32627833, PMID: 23395239). Given that such demographic shifts could influence treatment strategies, this topic would likely engage readers. If these temporal changes are potentially related to TDS, the authors should provide a literature-based discussion and offer their interpretation.
Reply 10
We thank the reviewer for this valuable suggestion. We have expanded the section on testicular cancer incorporating his suggestions.
Comment 11
The manuscript divides testicular tumors into germ cell tumors and sex cord stromal tumors. Although sex cord stromal tumors are much less common and the literature on them is limited, they may share biological mechanisms with TDS. It would strengthen the review if the authors could check whether any references discuss this possibility and briefly summarize their opinion on the potential overlap.
Reply 11
We thank the reviewer for this valuable suggestion. We have expanded the section on testicular cancer incorporating his suggestions.
Comment 12
The authors cite Reference 60 and discuss Disorders of Sex Development (DSD), congenital conditions involving atypical development of chromosomal, gonadal, or anatomical sex. Because DSDs partially overlap conceptually with hypospadias and cryptorchidism yet differ in other respects, a short literature-based explanation distinguishing DSDs closely related to TDS from those less related would help readers grasp the nuanced relationship between these entities.
Reply 12
Thank you for the suggestion, we added a comment of DSD in the introduction (lines 50-56).
Comment 13
Recent studies have gradually accumulated evidence suggesting that biological, anatomical, genetic, and environmental factors may contribute to gender identity. Particularly noteworthy are reports of transgender cases associated with cryptorchidism (PMID: 38249107). Although this is a highly sensitive topic and must be handled with care, if epidemiological or mechanistic evidence suggests any overlap between factors contributing to transgender identity and those underlying TDS, it would be worthwhile to mention this cautiously with appropriate references and interpretation.
Reply 13
We thank the reviewer for raising this sensitive issue. We have incorporated the limited information available in the literature into the text, limiting ourselves to observations based on scientific data rather than pure conjecture (lines 707-716).
Comment 14
Section 3, Maternal and fetal factors associated with TDS, provides a comprehensive discussion of maternal risk factors, but there is no mention of paternal risks. This omission could leave readers with an impression of imbalance. If relevant studies exist addressing paternal factors—such as the effects of paternal aging or environmental exposures—the authors should cite and discuss them, presenting their own perspective where possible.
Reply 14
We thank the reviewer for the important comment, we added the information regarding paternal factors potentially involved in TDS in section 5 (lines 617-634).
Reviewer 2 Report
Comments and Suggestions for Authors
Dear authors, after reading your manuscript, I have the following comments:
Your research does not have a clear, repeatable search strategy (for example, criteria for including or excluding people, search terms for databases). This means that the studies that were looked at may have been chosen in a way that makes them more likely to be biased, which could mean that important data is missed or that some findings are given too much weight. My advice is to convert to a Systematic Review or Meta-Analysis. This would necessitate the articulation of a transparent search strategy (including databases, keywords, and selection criteria) and the synthesis of the literature through quantitative or rigorously defined qualitative methodologies.
Allocate a section to proposed or validated mechanistic models that demonstrate the interaction of genetic variants (e.g., in AR, INSL3, NR5A1) with environmental disruptors and the resulting epigenetic modifications (e.g., AR promoter methylation). The article should aim to build a model that fits together rather than just listing factors. Introduce a conceptual diagram to visually represent the proposed multifactorial pathway of TDS, connecting maternal factors/EDCs to fetal gonadal dysfunction via genetic and epigenetic modifiers.
Author Response
Dear authors, after reading your manuscript, I have the following comments:
Comment 1
Your research does not have a clear, repeatable search strategy (for example, criteria for including or excluding people, search terms for databases). This means that the studies that were looked at may have been chosen in a way that makes them more likely to be biased, which could mean that important data is missed or that some findings are given too much weight. My advice is to convert to a Systematic Review or Meta-Analysis. This would necessitate the articulation of a transparent search strategy (including databases, keywords, and selection criteria) and the synthesis of the literature through quantitative or rigorously defined qualitative methodologies.
Reply 1
We appreciate the time and effort devoted by the reviewer to carefully reading our article and for his valuable suggestions, which have contributed significantly to improving our manuscript. Although the suggestion to transform our review into a systematic one is undoubtedly valid, we believe that the topic covered by our review can be adequately addressed using the narrative review method. However, we recognize that a description of the research strategy is essential to ensure the transparency and scientific rigor of our manuscript, so we have added a section dedicated to materials and methods in which we describe how we conducted our literature search.
Comment 2
Allocate a section to proposed or validated mechanistic models that demonstrate the interaction of genetic variants (e.g., in AR, INSL3, NR5A1) with environmental disruptors and the resulting epigenetic modifications (e.g., AR promoter methylation). The article should aim to build a model that fits together rather than just listing factors. Introduce a conceptual diagram to visually represent the proposed multifactorial pathway of TDS, connecting maternal factors/EDCs to fetal gonadal dysfunction via genetic and epigenetic modifiers.
Reply 2
Thank you for the suggestion, we added Figure 2 and the description of potential mechanistic model explaining TDS pathogenesis in section 8 (lines 792-815).
Round 2
Reviewer 1 Report
Comments and Suggestions for Authors
The authors have carefully and diligently addressed the previous comments, and I commend their persistence in improving the manuscript. Although there may still be room for deeper discussion—for example, regarding the interpretation of epidemiological trends in TDS, its increase in developed countries, and the relationship between rising age at testicular cancer diagnosis and TDS—the current revision already meets publication standards.
It is unusual to introduce new points at this stage, and I apologize for doing so. The following topics are optional, but the authors may consider them if they find them relevant:
-
Germ cell tumors and microRNA (PMID: 20332240)
-
MMP9, CSF1R, PTPRC in germ cell tumors (PMID: 30854095)
-
Testosterone/LH ratio; Reinke crystals (PMID: 28222406)
-
Sertoli cell dedifferentiation; TDS and gender identity reassignment (PMID: 24012888)
-
Testicular microlithiasis; KITLG, BMP7 (PMID: 30027931)
-
2D:4D digit ratio (PMID: 36789270)
Author Response
We would like to express our sincere thanks to the reviewer for his extremely valuable comments. Most of the studies suggested have been cited and included in the final version of the manuscript.